# Harnessing Machine Learning in Tackling Domestic Violence—An Integrative Review

**DOI:** 10.3390/ijerph20064984

**Published:** 2023-03-12

**Authors:** Vivian Hui, Rose E. Constantino, Young Ji Lee

**Affiliations:** 1Center for Smart Health, School of Nursing, The Hong Kong Polytechnic University, Hong Kong; 2Health and Community Systems, School of Nursing, University of Pittsburgh, Pittsburgh, PA 15260, USA; 3Department of Biomedical Informatics, School of Medicine, University of Pittsburgh, Pittsburgh, PA 15260, USA

**Keywords:** domestic violence, intimate partner violence, machine learning, big data, abuse

## Abstract

Domestic violence (DV) is a public health crisis that threatens both the mental and physical health of people. With the unprecedented surge in data available on the internet and electronic health record systems, leveraging machine learning (ML) to detect obscure changes and predict the likelihood of DV from digital text data is a promising area health science research. However, there is a paucity of research discussing and reviewing ML applications in DV research. Methods: We extracted 3588 articles from four databases. Twenty-two articles met the inclusion criteria. Results: Twelve articles used the supervised ML method, seven articles used the unsupervised ML method, and three articles applied both. Most studies were published in Australia (*n* = 6) and the United States (*n* = 4). Data sources included social media, professional notes, national databases, surveys, and newspapers. Random forest (*n* = 9), support vector machine (*n* = 8), and naïve Bayes (*n* = 7) were the top three algorithms, while the most used automatic algorithm for unsupervised ML in DV research was latent Dirichlet allocation (LDA) for topic modeling (*n* = 2). Eight types of outcomes were identified, while three purposes of ML and challenges were delineated and are discussed. Conclusions: Leveraging the ML method to tackle DV holds unprecedented potential, especially in classification, prediction, and exploration tasks, and particularly when using social media data. However, adoption challenges, data source issues, and lengthy data preparation times are the main bottlenecks in this context. To overcome those challenges, early ML algorithms have been developed and evaluated on DV clinical data.

## 1. Introduction

Domestic violence (DV) is a significant public health issue that threatens the physical and mental well-being of people around the world. According to the World Health Organization (WHO), strategies and advocacy campaigns to reduce DV have raised public awareness of the issue through media, including online platforms [1]. For example, the widespread use on social media of hashtags (i.e., a word or phrase denoted by a hash sign (#) to identify digital content on the same topic) such as #metoo, #notokay, #whyistayed, and #whyileft has aroused a whirlwind of online disclosure by victims and survivors, who have received emotional, informational, financial, and community support on various platforms such as Twitter, Facebook, and Reddit [2,3]. The DV experience is a sensitive and private topic, normally reserved to face-to-face conversations with healthcare providers and peers, but social media valuably serves as an anonymous platform for people to express their needs and feelings without the guilt or shame associated with traditional in-person interviews [4].

During the COVID-19 pandemic, in particular, the enormity of DV gained broad attention as victims were forced to isolate with their abusers under stay-at-home orders [5,6]. Traditional community centers, shelters, governmental buildings, or even social work services were paused or shut down during the pandemic. Victims turned to social media and other online platforms to express their emotions and concerns and seek help from others [7,8]. Interpreting digital content may offer a viable approach to identify potential victims, who can then be provided with social support. Analysis of social media content has been adopted in various fields to understand the patient population, such as in breast cancer, depression, and people with high suicide risk [9,10,11]. Extending that, social media could offer an important resource for clinicians, nurses, and researchers to understand the concerns and needs of DV victims and to guide further interventions.

Due to the large volume of text available from social media platforms, machine learning (ML) techniques have been widely adopted to process this wealth of data and analyze them systematically, thereby revealing patterns. ML is a method to learn and predict outcomes from patterns in data [12]. Over the past decade, ML studies have gained momentum in oncology, cardiology, and radiology. ML algorithms have been used in healthcare research across the continuum of early detection, diagnosis prediction, prognosis evaluation, and treatment [13,14,15,16]. They have been applied to various health data such as clinical notes, patient narratives, social media data, and patients’ free-text answers to questionnaires, with the goals of improving patient outcomes, deepening understanding of patients’ psychological conditions, and devising supportive management [17,18].

In this era of rapid technology advancement, there is a need to ground novel work in evidence-based research when advancing the field of ML in DV. However, to the best of our knowledge, current reviews have only examined technology-based interventions for violence against children [19], mental health detection [20], and concerning the challenges and opportunities of ML in general healthcare [21]. Although the application of ML to the DV population has increased over the past decade, no evidence summarizes the current state of science in ML applications in the DV field. Hence, the research questions were formulated as follows: 1. What types of ML have been applied in the DV field? 2. What data sources have been used for ML development and for what purposes in DV research? 3. What ML algorithms have been used? 4. What outcome variable has been assessed by ML? 5. What are the implementation challenges? An integrative review of ML applications in the DV arena will help clinicians and researchers understand how ML can be applied in the field, what data sources there are, what outcomes have been evaluated, and what challenges have commonly been encountered. This study will help clinicians and researchers design future ML applications by improving their understanding of the existing data sources and challenges facing ML applications, as well as generate clinical and policy implications. To those ends, this integrative review paper (1) reviews the current use of ML in DV research, and (2) identifies the challenges in implementing ML in DV research.

## 2. Materials and Methods

### 2.1. Research Design

In this study, domestic violence is defined as any type of violence that can happen in a household, including intimate partner violence (IPV), child abuse, and elder abuse. An integrative review is conducted from peer-reviewed scholarly articles on ML in DV research based on the framework from [22]. This review style is the broadest kind of method, covering different research designs, which enables us to fully understand the current state of the science of ML in DV.

### 2.2. Search Methods

We collected literature from PubMed, PsycINFO, CINAHL, Scopus, and Google Scholar on the application of machine learning in domestic violence research since this review paper is situated at the intersection of DV and machine learning. The rationale behind choosing Scopus and Google Scholar was to extract multidisciplinary research in the field via a natural language keyword search. Since we wanted to examine DV from social, psychological, and health-related perspectives, we decided to include PubMed, PsycINFO, and CINAHL as further search databases, for which we used controlled taxonomies. The key search terms were (“machine learning” OR “deep learning” OR “artificial intelligence” OR “AI” OR “supervised learning” OR “unsupervised learning”) AND (“domestic violence” OR “family violence” OR “intimate partner violence” OR “gender-based violence” OR “elder abuse” OR “child abuse”) (Table 1).

All types of supervised and unsupervised machine learning that are applied to DV were included. Supervised ML refers to building a model with a known goal of prediction or classification, while unsupervised ML works on feature extraction such as themes, topics, or sentiment [23]. This review was carried out in accordance with the Preferred Reporting Items for Systematic Review and Meta-Analysis (PRISMA) framework [24]. The inclusion criteria were use of the ML method to prevent, predict, assess, or explore DV, with the work published between 1 January 2010 and 1 January 2022. This study excluded any articles published before 2010, those neither applying the ML method nor applicable to the DV field, full texts other than in English, and review articles (Figure 1). The initial database of literature was collected and stored in Mendeley, a referencing software manager, to build a collection of scholarly articles. After data collection, we removed duplicates in the Mendeley software using its “duplicates removal” function, and subsequently performed manual screening to confirm the de-duplication results [25]. Abstract screening and full-text extraction were performed manually.

### 2.3. Quality Appraisal

To maintain internal validity and quality, two authors counterchecked the analysis and results of this study. The first author performed the initial screening based on the abstracts, created the tables and search strings, and synthesized the articles. The third author verified the screening process and counterchecked the result for each variable. Any disagreement was settled iteratively through discussion until the two researchers reached a consensus.

### 2.4. Data Abstraction and Synthesis

Data were evaluated and extracted from online databases with the full text available in English. Data retrieved from the articles included the ML method (i.e., supervised or unsupervised), study group, data source, use of ML, function, algorithm selection, outcome variable, and challenges, as shown in Table 2 and Table 3.

Supervised learning refers to the use of labeled datasets for classification or prediction of outcomes, while unsupervised learning refers to the use of unlabeled datasets for uncovering hidden patterns [47]. The form of learning was determined by the algorithm selected for training and evaluation in the literature. For the ML method, any studies that used supervised learning algorithms such as linear regression, logistic regression, decision trees, and support vector machine (SVM) were classified as supervised learning, whereas studies that used techniques such as natural language processing (NLP), clustering, and principal component analysis (PCA) were classified as unsupervised learning. The challenges in conducting and implementing ML in domestic violence studies are presented in Table 3.

## 3. Results

### 3.1. Overview

A total of 3588 articles were identified by the search strings used in Table 1. After the removal of duplicates (*n* = 2336), the remaining 1252 unique articles were screened based on the title and abstract; of those, 1092 articles were excluded as they were not about machine learning or domestic violence. The remaining 113 articles were assessed for eligibility, and 91 of the articles were removed after screening the full text, while 22 articles met the inclusion criteria (Figure 1).

Twelve articles used the supervised ML method, seven articles used the unsupervised ML method, and three articles applied a mixture of supervised and unsupervised techniques. There was often a DV population (*n* = 13) in the studies, followed by IPV (*n* = 5), gender-based violence (*n* = 1), child abuse (*n* = 2), mixed DV, and child abuse (*n* = 1). In terms of location, Australia (*n* = 6) and the United States (*n* = 4) were the main contributors of ML research in the DV research field, while the rest of the studies were conducted in the Netherlands, South Africa, Turkey, Sri Lanka, Peru, Taiwan, Spain, and China. Data sources for the articles ranged from social media (*n* = 12) to professional notes (*n* = 5), national databases (*n* = 3), a survey (*n* = 1), and the news (*n* = 1). Three types of ML usage among DV studies were identified, and illustrated in Figure 2: classification of the likelihood of DV (*n* = 8), prediction of future crime or DV (*n* = 7), and an exploration of themes or hidden topics (*n* = 8). The top three algorithms used for supervised ML training were random forest (*n* = 10), support vector machine (*n* = 8), and naïve Bayes (*n* = 7), while the most commonly used algorithm for unsupervised ML was latent Dirichlet allocation (LDA) for topic modeling (*n* = 2). Precision, recall, and the F1 measure were the most popular evaluation matrices for supervised learning studies (*n* = 13). There were eight types of outcomes assessed among the 22 articles: mental and emotional health (*n* = 3), substance-related problems in child abuse (*n* = 2), offenders and recidivism-related (*n* = 2), victimization or revictimization (*n* = 5), facial injuries (*n* = 1), reasons for staying/leaving relationship (*n* = 1), building a classifier/framework for DV and CA cases (*n* = 4), and themes and topics (*n* = 4). Of the 22 articles, 16 were found to be relevant to the challenges of conducting and implementing ML in DV research.

### 3.2. The Current Use of ML in DV Research and the Outcome of the Study

In terms of the current use of ML in DV research, three purposes were identified: classification (*n* = 8), prediction (*n* = 7), and exploration (*n* = 8). As DV crisis support groups have thrived on social media and formed online communities recently, identification of critical posts requires manual browsing to classify people at high risk of DV, which is time-consuming and ineffective with such a large amount of data. Recent research published by Subramani et al. (2018) [44] delineated the steps needed to apply supervised ML to classify critical posts (i.e., posts with emergency needs or red flags on DV) from a Facebook DV support group. Subramani et al. (2019) [42] also published another study on multi-class identification (i.e., identifying multiple classes of terms with the most common words) from domestic violence online posts on Twitter. They constructed the novel “gold standard” dataset from the social media DV crisis support group with multi-class annotation (i.e., manually annotate postings and train the ML model) using supervised ML techniques.

In terms of prediction, Berk et al. (2016) applied supervised ML to forecast the future dangerousness of offenders in over 18,000 arraignment cases from a metropolitan area in which the offender faced DV charges [28]. Another crime prediction study on DV was conducted by Wijenayake et al. (2018) [45]. Their team employed the decision tree approach to predict recidivism among DV offenders. Their results achieved a good prediction under the ROC area with only three features and four leaf nodes, which supported an effective prediction model of recidivism among DV cases.

Apart from crime detection and forecasting, exploratory research is abundant on DV in online forums such as Sina Weibo, Reddit, and Twitter. Liu et al. (2021) [34] explored the short-term outcomes of DV on mental health by collecting data from Sina Weibo. They utilized unsupervised ML to identify an individual’s mental health status. Similar approaches have been applied on Reddit to understand the online disclosure of DV and develop classifiers for “abuse” and “non-abuse” posts [39]. Moreover, Xue et al. (2019) [46] used topic modeling techniques on Twitter to understand the DV topics among Twitter users. They used unsupervised ML techniques to identify topics that appear most frequently. Overall, the social media online disclosure of DV has been examined, covering general topics, emergency classification, and mental health state classification.

### 3.3. Challenges in Conducting and Implementing ML in DV Research

#### 3.3.1. Limited Availability of Data Sources

One of the main challenges to implementing ML in DV has been the data source issue (*n* = 9), including generalizability, reliability, and availability. Studies using a national database (*n* = 3) are mostly cross-sectional surveys, and self-reported responses lead to recall bias or lapse of time issues, which may affect the accuracy of the prediction. As discussed by Berk et al. (2016) [28], limited electronic data and court documents for DV cases can be extracted for ML algorithms’ development. Some jurisdictions’ records are still in written form instead of electronic. In terms of social media data, Liu et al. (2021) [34] illustrated that the user’s information on posts can be inaccurate, and that only incomplete demographic data can be captured from online posts (Table 3).

#### 3.3.2. Excessive Data Collection, Annotation, and Model Interpretation Time

Eight studies reported that the excessive time spent on collecting, annotating, and interpreting data is another critical challenge to conducting an ML study in DV research. Subramani and colleagues highlighted the difficulties for annotators in differentiating the categories [40,41,42,43,44], including the laborious annotation process and agreement between annotators. Similar comments were also made by Liu et al. (2021) [34] regarding unavoidable subjectivity when different judges were judging the data. Social media data contain many daily conversations, which generate much noise. For example, Chu et al. (2020) [29] reported that frequent use of words such as “mother” and “abuse” can confuse researchers, making it hard for them to differentiate between topics, while Homan et al. (2020) [32] also mentioned that noise from spam bots, along with jokes or mistruths from users, are not easily filtered by the text-mining method.

#### 3.3.3. Difficulties in Technology Adoption

In terms of the implementation, four articles (*n* = 4) were included [26,28,33,37], and three issues were identified: end users’ acceptance (*n* = 1) [26], professional compliance (*n* = 2) [28,33], and professional training (*n* = 1) [37]. As discussed by Amrit et al. (2017) [26], end users (i.e., nurses and physicians) have limited understanding of the model development or how to apply a prototype (i.e., early product sample) in their work; providing further information in these regards is important to increase the acceptance by end users.

Apart from end users’ acceptance, compliance from professionals such as police officers and magistrates was discussed by Karystianis et al. (2020) [33] and Berk et al. (2016) [28], respectively. The leniency of magistrates and non-systematic guidelines for how police officers should take records contributed to the unreported DV cases. Therefore, professional training is critically important to implement ML in DV research. In that spirit, Perron et al. (2019) [37] highlighted the importance of adding scientific training and data science skillsets to the curriculum for social science and healthcare professionals.

## 4. Discussion

Over the past decade, ML has been used to classify high-risk victims, predict future violent episodes, and explore the hidden themes from victims and survivors among different types of data sources through both supervised learning and unsupervised learning. Given that this is the first wave of ML research in DV, several challenges have been identified, overcoming which may facilitate future research.

### 4.1. Current Use of ML in DV

#### 4.1.1. Supervised Method

Leveraging ML to predict the likelihood of DV and uncover the hidden themes or patterns from DV are the main goals identified in our review. We found that supervised ML dominates in the development of classification models to predict the future occurrence of DV, re-victimization, and recidivism. This trend can be explained by the difficulties in detecting DV in advance [48,49]. Traditionally, detecting victims has relied heavily on self-reports from victims or reports from healthcare providers after assessing injuries in the emergency room [48,50]. Early detection and screening of DV are time-consuming because those methods involve long-term observation through home visits by social workers, wound nurses, or community nurses. Usually, it is too late to intervene when healthcare professionals identify a suspected DV case from their work. Leveraging ML for DV interventions is not just limited to early crisis identification and classification from digital documents, but can also be extended to legal arraignments and decisions. Apart from assisting healthcare professionals to make clinical decisions, the use of ML in DV research has also improved legal judgment-making from the judicial perspective. Therefore, using ML to classify the level of danger of DV can potentially improve screening efficiency and increase the early detection rate to avoid severe physical injuries or further victimization from DV exposure.

#### 4.1.2. Unsupervised Method

Exploring the themes and topic patterns from DV survivors using unsupervised ML (*n* = 5) was another trend found in our study. Over the past decade, social media has become a mainstream source of news delivery, promotion, and information exchange, where DV survivors can express their feelings and stories to draw public attention to DV anonymously [3]. It has also provided a dynamic source of support for victims, who can seek help and support from each other. Simultaneously, many non-governmental organizations and activists have launched online health promotion campaigns on DV prevention by using hashtags such as #metoo, #notokay, and #maybehedoesnthurtyou [51,52,53]. Accompanying these trends, many DV stories have been shared and forwarded globally over social media, allowing researchers to explore firsthand stories through text-mining techniques such as topic modeling and sentiment analysis. Since DV cases are underreported, social media sheds light on the need to create more sources of DV information that are beneficial to early detection, prevention, and intervention.

Our study also found that digital professional notes were used as resources for unsupervised ML. With the popularity of digitizing documents for efficient management, different professions are now transforming their written notes into digital notes [54]. In our review, the digital notes included consultation notes [26], electronic records of arraignments [28], police records [33], child welfare agency written summaries [37], and reoffending records [45]. Most of the data sources were from professions that commonly deal with DV such as law, disciplinary forces, social work, and counseling. Surprisingly, our review did not find any data sources from electronic health records (I) or DV shelter databases. One of the plausible reasons is they may contain too much information or they may be restricted by privacy considerations and hospital security policies. The data preprocessing time for clinical EHR data is relatively long and the procedure to request data that contain DV information is complicated due to restricted data availability across institutions. Some records, however, such as wound assessments from emergency rooms, or outpatient clinic social worker assessments that specifically capture the recent injury and family functioning, may arise as sources of data for future data-mining studies. Additionally, traditional qualitative studies in DV mainly recruited survivors from shelters for convenience and privacy protection, which may have limited the number of participants. We expect that more online narratives will become available from shelter websites or support group forums under the trend of self-disclosure encouraged by social media.

#### 4.1.3. Current Outcomes Evaluated by ML in DV Research

In terms of the outcome variables evaluated, our review identified eight types of variables. Building an ML classifier to predict DV cases is the dominant outcome variable. One of the reasons is that screening and early detection of DV have posed a challenging task for decades, and leveraging ML techniques could potentially identify more victims at risk with minimal manpower and time. As DV victims have higher odds of mental health disorders [55], extracting wordings and features for mental or emotional health prediction is reasonable because ML allows researchers to cluster the patterns and identify possible features of multidimensional data. Our review has shown that current outcome variables also focus on a psychosocial aspect, such as the reasons for staying in or leaving a relationship, and legal aspect, such as offenders and recidivism-related issues. This finding is consistent with previous literature regarding the needs and safety concerns of DV victims, where legal help is of the highest priority [56].

### 4.2. Challenges in Conducting and Implementing ML in DV

An essential issue reported around conducting ML in DV research is the lack of data sources. The availability of electronic records for ML building is a critical component. As reported by Berk et al. (2016) [28], some records are still in a paper format. In certain rural areas, some clinicians and consultants still use a paper format for filing storage [57]. The scanning of documents is labor-intensive, and text identified from handwriting can be inaccurate, which can impede the use of text-mining techniques to retrieve the data. This result aligns with a recent review illustrating how to standardize clinical processes and integrate fragmented records for ML training in healthcare [21]. Furthermore, there are noises and response biases in patient-reported data, especially on social media [58]. Social media research mainly relies on users’ posts, and self-reported DV experiences may undermine the reliability and accuracy of real situations. Social media research data should only comprise those that are public, all the posts should be anonymized, and personal identifiers such as specific geographic locations should be removed from the dataset.

We also observed that current study populations were largely limited to DV and IPV, while child abuse and elder abuse were rarely selected for ML. This may be explained by the lack of self-reported child abuse and elder abuse cases on the internet or in digitized notes. Besides the fact that DV and IPV are more commonly reported by healthcare professionals when compared to elder abuse [59], there is also the fact to consider that children and the elderly have not traditionally been likely to share their abuse experiences in person or through online disclosure. However, with increasing computer literacy and the prevalence of technology use, Haris (2014) [60] and Dilci (2019) [61] suggested that the elderly and teenagers may increasingly use social media or online chat box systems to disclose their experiences. The rise of social media influencers or key opinion leaders can contribute to self-disclosure behavior among teenagers [62]. A recent study showed that there was a significant increase in child abuse and DV content generated on Twitter by children during the COVID-19 pandemic [63]. Moreover, since working from home became widely accepted and adopted, violence against family members appeared as an escalating threat to public health [64]. Although the COVID-19 pandemic is thought to have passed its peak, it is expected that online platforms will continue to grow and become the main source of self-disclosure.

Currently, ML is mainly focused on predicting future abuse and uncovering underlying risks, meaning, information needs, and mental health consequences from DV survivors. Attention to the psychological status of DV survivors has only examined short-term mental health outcomes such as stress and depression [34], and future research efforts can be devoted to promoting positive mental change following the traumatic experience of DV. Across the variables covered in our review, few ML studies identified positive mental health outcomes (e.g., resilience and personal growth). One of the reasons for this could be the challenges of measuring the psychological status from big data, which requires a systematic way to interpret the findings, such as an ontology or a dictionary. Similar to the findings from Lee et al. (2020), we note that utilizing social media data to extract emotional needs via ML techniques is challenging [65]. Compared to negative outcomes, positive mental health concepts usually have obscure and inconsistent linguistic features that are not yet standardized. Further studies are required to develop a pipeline to structure and extract this psychological information from the various available data resources. Additionally, using ML models to examine psychological traits, states, and changes can be beneficial for advancing health science research and monitoring the mental health status of patients. However, it is commonly warned that improper use of predicted results may constitute a threat to the well-being of people.

In addition to the challenges of psychological prediction, leveraging ML techniques in DV research also suffers from limitations. Only two studies found that ML techniques were useful and practical when applied to real-world settings [29,32]. While a model may appear successful in classifying the correct groups and needs, deployment in real-world settings throws up different challenges. For example, computer systems and formats differ between medical institutions and clinics. Aggregation of clinical data across institutions to address semantic interoperability but without clear guidelines and consent from the concerned parties may trigger privacy issues and ethical concerns, as well as posing a risk of data leakage [66]. In addition, the ethics of collecting, storing, and sharing mental health data present another significant obstacle in ML research, as do the degrees of privacy and autonomy afforded to ML systems.

## 5. Implications

Our study revealed the discrepancies between law enforcement and unsystematic reporting of DV cases. Training among magistrates on how to standardize the reporting procedure in a digital format could offer a chance to remove any leniency and improve the accuracy of ML prediction. In addition, our study showed that there is acceptance by end users in different professions, ranging from police officers to legal magistrates and healthcare professionals. However, we are uncertain whether ML techniques are clinically meaningful and practically useful in reality. It is suggested that including DV domain experts from in the multidisciplinary team, to provide feedback and monitor ML implementation, could be helpful to improve the adoption rate of this technology in clinical settings. For example, the prediction results from ML models can be reviewed by DV domain experts (i.e., medical social workers, nurses in emergency room, community nurses, and lawyers), who compare the results with manual screening to check for sensitivity and specificity qualitatively. In addition, future research could examine the level of technology adoption after implementation and validate the accuracy of ML models in different settings.

Using ML as a method to curate data from digital text is promising in DV research. Most of the current reviewed publications were produced by researchers from the computer science and social sciences fields. Increased training for DV researchers regarding the use of ML and what algorithms are available for them to analyze their data could facilitate greater ML research from healthcare perspectives. For practitioners, ML techniques are impactful for real-life applications in clinics or community centers to identify DV victims from a digital database. The development of cross-reporting systems between multiple community centers or clinics, with ML functions to screen victims who are seeking help online, could better identify DV perpetrators and potential victims in the community. However, the results generated from ML should be interpreted with caution as the current state of the science in ML research uses diverse datasets, which limit the applicability of the outcomes. In addition, considering that the prediction accuracy of ML algorithms is not compared against the usual traditional screening methods, practitioners working with DV populations are advised to interpret ML results with critical judgment based on their clinical experience.

## 6. Limitations

There are several limitations to this study. First, the scope of this review covers the broad spectrum of ML applications in DV for any country. In the field of nursing and public health, healthcare settings differ, the practices and systems for supporting DV survivors vary tremendously, and so do the characteristics of DV. Accordingly, the use of ML in DV research may not be the same internationally. Therefore, it may not be possible to generalize the conclusions of our study to all countries. While we summarized the challenges of ML application in DV based on the limitations mentioned in each included study, other variables such as the setting, target population, data source, and use of ML were not compared. Hence, future studies considering the challenges of ML in DV research should regard our findings with caution.

## 7. Conclusions

ML research in DV has advanced exponentially in the past decade. This review provides direction for researchers and clinicians on how ML has been deployed in the field of DV for the classification, prediction, and exploration of DV topics, and identifies the challenges of technology adoption, data source availability, and lengthy data preparation, which were highlighted in previous literature. The development of ML algorithms is still in its infancy within clinical settings due to restricted interoperability between institutions. With ML tools and research funding becoming more accessible, it is expected that ML in DV research will keep expanding, and more innovative strategies will be explored shortly.

## Figures and Tables

**Figure 1 ijerph-20-04984-f001:**
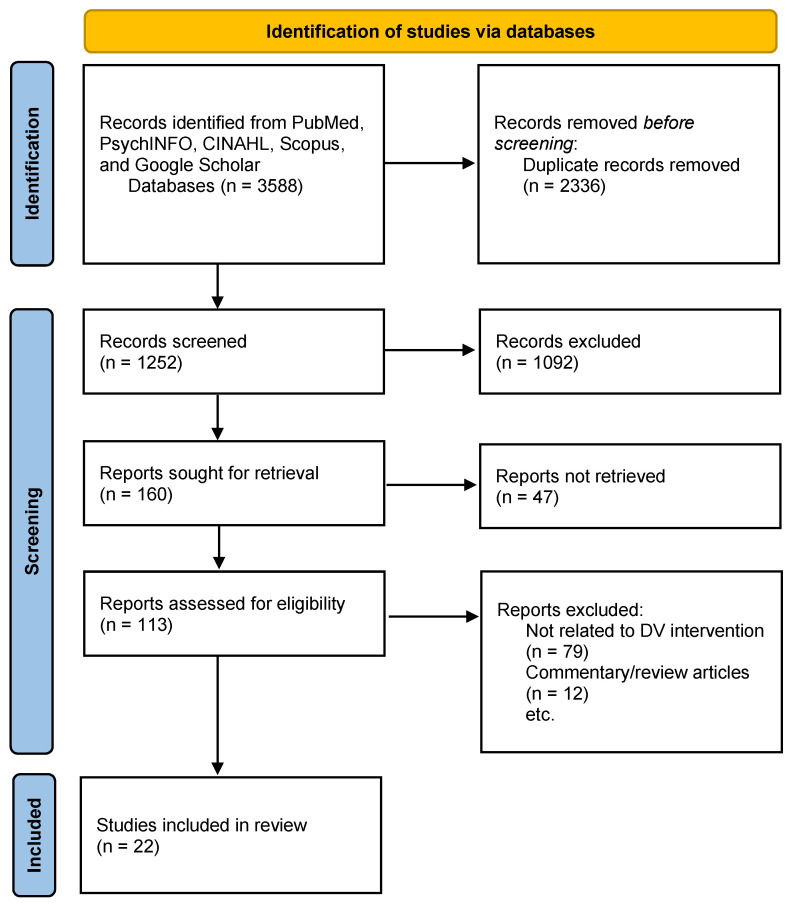
PRISMA Flow chart—integrative review of machine learning in domestic violence.

**Figure 2 ijerph-20-04984-f002:**
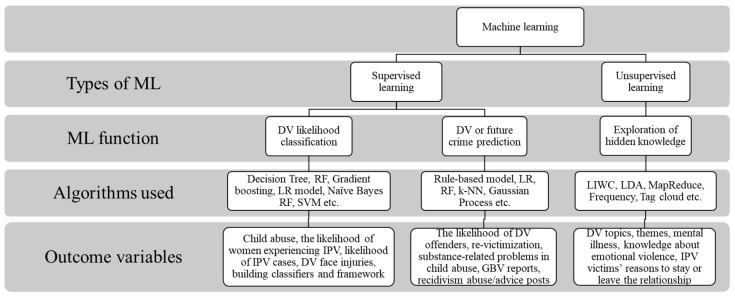
Technicalities of ML application in DV research. Note: SVM: Supper vector machine, RF: Random forest, LR: Linear regression, LDA: Latent Dirichlet Allocation, CNNs: Convolutional neural network, RNNs: Recurrent neural network, LIWC: Linguistic Inquiry Work Count, k-NN: k-nearest neighbors.

**Table 1 ijerph-20-04984-t001:** Search strings for different search engines.

Search Engine	Search Strings
** PubMed **	((“machine learning”[Mesh] OR “supervised machine learning”[Mesh] OR “deep learning”[Mesh] OR “unsupervised machine learning”[Mesh] OR artificial intelligence[tiab] OR Twitter[tiab] OR social media[tiab]) AND (“intimate partner violence”[Mesh] OR “gender-based violence”[Mesh] OR “domestic violence”[Mesh] OR intimate partner violence[tiab] OR domestic violence[tiab] OR child abuse[tiab] OR spouse abuse[tiab]))
** PsycINFO **	((machine learning/OR supervised machine learning/OR unsupervised learning/OR artificial intelligence*.ti,ab. OR (Reddit or Facebook or Twitter or social media). ti,ab.) AND (domestic violence/OR intimate partner violence/OR child abuse/OR elder abuse))
** CINAHL **	(TI machine learning* OR AB machine learning* OR TI supervised learning OR AB supervised learning OR TI unsupervised learning* OR AB unsupervised learning OR MH “social media”) AND (TI domestic violence OR AB domestic violence OR TI domestic abuse OR AB domestic abuse OR TI intimate partner violence OR AB intimate partner violence OR TI spouse abuse OR AB spouse abuse OR TI child abuse OR AB child abuse OR MH “domestic violence+”)) NOT PT dissertation
** Scopus **	(“domestic violence” OR “domestic abuse” OR “intimate partner violence” OR “spouse abuse” OR “child abuse” OR “partner abuse” OR “gender-based violence”) (“mobile application*” OR app OR smartphone* OR Facebook OR Twitter OR “social media” OR “cell*phone*” OR “text message*” OR “smartphone*” OR “crowdsourcing” OR “online service*” OR “social media”)
** Google Scholar **	“machine learning” OR “deep learning” AND “domestic violence” OR “intimate partner violence” OR “elder abuse” OR “child abuse” OR “gender-based violence” AND “classification” AND “prediction”

**Table 2 ijerph-20-04984-t002:** An integrative review of the included studies (results shown in alphabetical order).

Author (Year)	ML Method	Study Group	Country ^1^	Data Source	Use of ML	ML Function	Algorithm Selection		Outcome Variable
							Training	Evaluation	
Amrit et al. (2017) [26]	S and US	Child abuse	Netherlands	Consultation notes	To predict whether a child suffers from abuse using classification models; To implement decision-support API.	Classification	Naïve Bayes RF, SVM	Precision, Accuracy, Recall, F1, ROC Curve	Child abuse case
Amusa et al. (2020) [27]	S	IPV	South Africa	Survey	To classify women based on their likelihood of experiencing IPV.	Classification	Decision Tree, RF, Gradient Boosting, LR Model	Precision, Accuracy, Recall, F1, ROC Curve	The likelihood of women experiencing IPV
Berk et al. (2016) [28]	S	DV	U.S.	Electronic records of arraignments	To forecast the future dangerousness of offenders in over 18,000 arraignment cases from a metropolitan area in which the offender faces DV charges.	Prediction	RF	Confusion Matrix	The likelihood of DV offenders
Chu et al. (2020) [29]	S and US	IPV	China	*Baidu Tieba’s* IPV Group	To understand themes for emotional and informational support from IPV by analyzing with automatic content analytics.	Exploration	k-NN, Naïve Bayes, LR, LDA	Accuracy, F1	Themes for emotional and information support of IPV
Guerrero (2020) [30]	S	IPV	Peru	Registered denouncements	To compare the classifier models in order to predict IPV.	Classification	LR, RF, SVM, Naïve Bayes	Precision, Accuracy, Recall, F1	The likelihood of IPV cases
Hsieh et al. (2018) [31]	S	IPV	Taiwan	IPV report form and danger assessment form	To build a repeat victimization risk prediction model.	Prediction	RF	Accuracy, F1	The likelihood of re-victimization
Homan et al. (2020) [32]	S	IPV	U.S.	Twitter	To analyze social media data for the reasons victims give for staying in or leaving abusive relationships.	Exploration	Naïve Bayes, Linear SVM, Radial Basis Function	Accuracy, Confidence Score	IPV victim’s reasons to stay or leave the relationship
Karystianis et al. (2020) [33]	US	DV	Australia	Electronic police records	To present the prevalence of extracted mental illness mentions for persons of interest (POIs) and victims in police-recorded DV events.	Exploration	GATE Text Engineering, International Classification of Diseases (ICD-10)	Precision	DV victims’ mental illness
Liu et al. (2021) [34]	US	DV	China	Weibo	To explore the short-term outcomes of DV for individuals’ mental health.	Exploration	Linguistic Inquiry and Word Count	Pearson’s Correlation Coefficient	DV mental health short-term outcome
Majumdar et al. (2018) [35]	S	DV	India	News (or social media)	To develop a DV face database and a deep learning framework for detecting injuries.	Classification	SVM, k-NN, Naïve Bayes, Random Decision Forest (RDF)	Accuracy	DV face injuries
Özyirmidokuz et al. (2014) [36]	S	DV	Turkey	(TURKSTAT) website	To extract meaningful knowledge from the “emotional violence against women” dataset.	Exploration	Decision Tree	Accuracy, Cross-Validation	Knowledge about emotional violence in DV
Perron et al. (2019) [37]	S and US	Child abuse	U.S.	Child welfare agencies wrote summaries	To better understand and detect substance-related problems among families investigated for abuse or neglect.	Prediction	Rule-Based Model, LR, RF	Global Accuracy, Sensitivity, Specificity	The likelihood of substance-related problems in child abuse
Rodríguez-Rodríguez et al. (2020) [38]	S	GBV	Spain	Spanish national database	To forecast the reports and complaints of GBV.	Prediction	LR, RF, k-NN, Gaussian Process	Accuracy from Root Mean Squared Error and Standard Deviation	The likelihood of GBV reports
Schrading et al. (2015) [39]	S	DV	U.S.	Reddit	To develop classifiers to detect submissions discussing domestic abuse.	Classification	Perceptron, Naïve Bayes, LR, RF, Radial Basis Function SVM, linear SVM	Confusion Matrix, 10-Fold Cross-Validation	Building a classifier of DV content in the post
Subramani et al. (2018) [40]	US	DV	Australia	Twitter	To discover the various themes related to DV.	Exploration	MapReduce, Frequency, Tag Cloud	Precision, Recall, F1	Themes related to DV
Subramani et al. (2018) [41]	S	CA and DV	Australia	Facebook	To develop a framework to identify CA and DV posts from social media automatically.	Classification	Linguistic Inquiry and Word Count, Bag of Words, SVM, Decision Tree, k-NN	Precision, Recall, F1, Accuracy	Building framework of CA and DV content in the post
Subramani et al. (2019) [42]	S	DV	Australia	Twitter	To classify DV online posts on Twitter.	Classification	CNNs, RNNs, LSTMs, GRUs, BLSTMs, RF, SVM, LR, Decision Tree	Precision, Recall, F1, Accuracy	Building a classifier of DV content in the post
Subramani et al. (2017) [43]	S	DV	Australia	Facebook	To predict the accuracy of the classifiers between abuse or advice discourse.	Prediction	SVM, Naïve Bayes, Decision Tree, k-NN	Precision, Recall, F1, Accuracy	The likelihood of abuse or advice based on the post
Subramani et al. (2018) [44]	S	DV	Australia	Facebook	To identify DV victims in critical need automatically.	Classification	CNNs, RNNs, LSTMs, GRUs, BLSTMs	Precision, Recall, F1, Accuracy	Building a classifier for DV critical cases
Wijenayake et al. (2018) [45]	S	DV	Sri Lanka	Re-offending database	To predict recidivism in DV.	Prediction	Decision Tree	ROC Curve	The likelihood of recidivism in DV
Xue et al. (2019) [46]	US	DV	Canada	Twitter	To examine DV topics on Twitter.	Exploration	LDA		DV-related topics
Xue et al. (2020) [8]	US	DV	Canada	Twitter	To examine the hidden pattern of DV during COVID-19.	Exploration	LDA		DV-related topics

Note: Literature is shown in alphabetical order. S: Supervised, US: Unsupervised, NLP: Natural language processing, C: Classification, IPV: Intimate partner violence, GBV: Gender-based violence, CA: Child abuse, SVM: Supper vector machine, RF: Random forest, LR: Linear regression, LDA: Latent Dirichlet allocation, ROC: Receiver operating characteristic, CNNs: Convolutional neural network, RNNs: Recurrent neural network, LSTMs: Long short-term memory neural networks, GRUs: Gated recurrent units, BLSTMs: Bidirectional long short-term memory networks, k-NN: k-nearest neighbors. Note. ^1^. Country was determined by affiliated institutions of the first author in each article.

**Table 3 ijerph-20-04984-t003:** Challenges in implementing machine learning in domestic violence.

Author (Year)	Challenges Mentioned
Amrit et al. (2017) [26]	▪ Moral and ethical challenges ▪ Limited understandability of the model for the end users ▪ Having a prototype based on the prediction model accepted by the end users ▪ Suggestion: option for a professional to provide the model with feedback
Amusa et al. (2020) [27]	▪ Data sources from a cross-sectional survey cannot guarantee accurate predictions due to self-reported responses, recall bias, etc. ▪ The decision tree model is not robust to bias resulting from other essential risk factors that were not included in the model
Berk et al. (2016) [28]	▪ The leniency of the magistrates may potentially lead to more unreported DV cases that are difficult to manage without a court order during the post-arraignment period ▪ Limited electronic data availability for DV cases and court documents ▪ The difficulties of obtaining information from jurisdictions in electronic form
Hsieh et al. (2018) [31]	▪ Using a survey may have a lapse of time, and the victim’s negative perception toward the violent experience may be reduced over time, which may affect the accuracy of the machine learning model prediction
Chu et al. (2020) [29]	▪ Frequent appearance of words such as “mother”, “abuse”, and “perpetrator” makes it hard to differentiate between topics (too much noise for topic modeling) ▪ Generalizability
Guerrero (2020) [30]	N/A
Homan et al. (2020) [32]	▪ Bias toward female victims and male abusers ▪ Unique or rare forms of abuse missing ▪ Noise (spam bots, lies by the users, jokes that were missed by filters)
Karystianis et al. (2020) [33]	▪ The police officer’s training on mental health and DV is limited ▪ The compliance from the police officers with taking records
Liu et al. (2021) [34]	▪ User information on posts can be inaccurate (fake profile and reports) ▪ Unavoidable subjectivity when judging the posts even with multiple judges ▪ Demographic information is incomplete on social media
Majumdar et al. (2018) [35]	N/A
Özyirmidokuz et al. (2014) [36]	N/A
Perron et al. (2019) [37]	▪ The number of documents that were manually coded was determined almost exclusively by available financial resources ▪ The core training curriculum in the applied social sciences needs to consider the variety of skills and knowledge needed to maximize the value of different types of data (scientific training is important)
Rodríguez-Rodríguez et al. (2020) [38]	N/A
Schrading et al. (2015) [39]	▪ Annotators had a hard time differentiating the categories
Subramani et al. (2018) [40]	▪ Domain experts from DV are needed to analyze the various health problems that were extracted
Subramani et al. (2018) [41]	N/A
Subramani et al. (2019) [42]	▪ The annotated corpora in the DV domain are important for ML ▪ The annotation process is tedious and laborious
Subramani et al. (2017) [43]	▪ Informal language use in short-text messages creates ambiguity ▪ The sparsity of instances of specific intent classes creates data imbalance
Subramani et al. (2018) [44]	▪ Laborious annotation process
Wijenayake et al. (2018) [45]	N/A
Xue et al. (2019) [46]	▪ No information about the gender or demographic information on Twitter users, which limits the generalization of the study findings to the general population
Xue et al. (2020) [8]	▪ Twitter data cannot represent the entire population’s opinions ▪ The search terms used in the study mostly reflect the terminology used by professionals rather than victims when discussing DV

## Data Availability

Data supporting reported results can be obtained on request.

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
