# Peer review of "Harnessing Machine Learning in Tackling Domestic Violence—An Integrative Review"

_ijerph, 2023, doi:10.3390/ijerph20064984_

Round 1

Reviewer 1 Report

1. The authors should further clarify the purpose and rationale behind their review. While a systematic and synthesized approach is beneficial, it alone is not sufficient justification for a review of this scope. The authors should provide a deeper explanation for the need and significance of their review

2. The overall contributions and benefits of this paper in the field of domestic violence, deep learning/machine learning should be emphasized more prominently.

3. The review tends to focus on surface-level applications of deep learning/machine learning in the context of domestic violence. However, there are deeper issues related to this topic and deep learning/machine learning in general that could be explored in more depth.

4. The authors should address some shortcomings in their paper. The background of deep learning/machine learning should be robust enough to assist other readers in understanding these concepts. The technicalities of deep learning/machine learning should be illustrated with appropriate visualizations to make it more accessible to readers outside of this research field. This approach will also attract more readers to appreciate the authors' work.

5. The authors should include more information on the selection of indexing databases used in the paper and explain their reasoning. Additionally, they should also mention why certain databases, such as Web of Science, which is known for its highly prestigious research, were not included.

6. Table 2 is lacking in detail, only providing a general summary. The authors should take advantage of the space and this opportunity to extract the most valuable information from these papers and provide their own original insights in a scientific manner. Merely mentioning training, evaluation, etc. does not add much value.

7. The authors should explain their rationale for selecting the papers listed in Table 3. They should also provide their own insights and scientific commentary on the challenges and advantages of the papers to assist other experts in advancing the field further.

8. The authors should recognize and examine other existing review papers on the same topic.

9. The paper contains a significant amount of ambiguity, particularly for those not familiar with the field of DL/ML. The paper should be more accessible to readers without prior background in the field in order to fully appreciate the review.

10. Given the current number of papers reviewed, this review paper appears to be lacking as many of the claims are not sufficiently supported by references. The authors may add more.

11. Can the authors please provide a more detailed, statistical, mathematical or appropriate explanation for the method used to eliminate duplicates?

12. Searching methods should have better reasoning and broader technicalities. Addition of references may also help with why the authors had this type of approach towards their aim.

Author Response

Dear reviewer, 
Thank you for your valuable comments. Please see the attached file for the point-by-point revisions. Thank you again for your time. 

First author of manuscript ID: ijerph-2187063

Reviewer 2 Report

1.     From this research I am unclear about the inclusion criteria for selecting the articles for this research and how they are satisfied with the selected number of measures, need to be clearer on this issue. Have they used any statistical sampling techniques also?

2.      I don’t see the most recent articles included in this study, the authors need to include more recent studies such as for the years 2022 and 2021 with a proper logical sequence.

3.     I am not clear about the significance of this study compared to existing studies, please clarify it.

4.    I have also seen that; they have selected articles from different countries and regions. But the pattern of domestic violence in different regions is different. For understanding easily for the reader, the author can rearrange their study according to the regions.

5.  I think there is a miss matching of the page number, just observe it and make a correction to it

Author Response

Dear reviewer, 

Thank you very much for your valuable comments. Please see the attached point-by-point response.

Yours sincerely, 
First author of the manuscript ID: ijerph-2187063

Reviewer 3 Report

Article titled “Harnessing machine learning in domestic violence – an integrative review”

The following manuscript is well managed but not depicted the complete story from objectives to conclusion. The current article looks like meta-analysis and bibliometric analysis. See the literature and add more about your set one dimension or direction.

The Introduction should be rewritten and to the point. Basically, the Introduction should clearly state the research questions and identify the gap of the research. Why is the topic important (or why do you study on it)? What are research objectives? What has been studied? What are your contributions? What is the novelty of this research?

There is not any framework mentioned for inclusion and exclusion criteria like PRISMMA framework etc.

The title of the research should be rewrite because it’s depicted something else about ML. there are a lot of minor grammatical mistakes.

Visualization and descriptive should be added in the document.

Abstract should be revised because it is not giving a whole message about the whole article.

Table 2 is well managed to compare the related articles.

Author Response

(The authors gave the same response as above.)

Round 2

Reviewer 2 Report

Thank you all for explaining the queries mentioned in the first review.

Author Response

Dear Reviewer,
We greatly appreciate your time. Thank you very much. 

Authors

Reviewer 3 Report

Good work to improve your manuscript in the next level. I have found something which is not incorporated yet.

Please revise the suggested changes to improve the following article.

1. Please follow the actual PRISMA framework (in MS word format) by using links below with inclusion and exclusion criteria.

a. https://guides.lib.unc.edu/prisma

or

b. https://guelphhumber.libguides.com/c.php?g=213266&p=1406923

2. Table 2 and PRISMA framework Diagram have the same purpose. Please use one of them and further explain in the text.

3. As written in article "A total of 3588 articles were identified by the search strings used in Table 1. After the removal of the duplicates (n=2336)"

These text should reflect through PRISMA framework.

4.  PRISMA framework diagram should be a part of methodology, not in the results section.

5. What is a new in Figure 2 and give reference. Use the same color if required to read within the following flow chart/diagram.

6. In Abstract Conclusions:

"Leveraging the ML method to examine DV has advanced exponentially in the past decade. However, further research is needed to examine the adoption difficulties in different settings and compare the ML effectiveness with the traditional methods"

It's not enough and concluding your whole article.

7. Don't use same text for abstract and conclusion. Like "ML research about DV has advanced exponentially in the past decade."

8. Please conclude as per the literature says, not as per the author(s) understanding. Like gap of research  must be address in the conclusion and abstract.

(this integrative review paper aims to 1) review the current use of ML in DV research, and 2) identify the challenges in implementing ML in DV research).

Author Response

Dear reviewer, 
We appreciate your time and suggestions. Please see the attachment. Thank you very much again. 

Authors
